# Digital transition in rural emergency medicine: Impact of job satisfaction and workload on communication and technology acceptance

Joachim P. Hasebrook[1]*, Leonie Michalak[2], Dorothea Kohnen[3],
Bibiana Metelmann[4], Camilla Metelmann[4], Peter Brinkrolf[4], Steffen Flessa[5],
Klaus Hahnenkamp[4]

**1** Steinbeis University, Berlin, Germany, **2** Curacon Ltd., Muenster, Germany, **3** University Leuven, Leuven, Belgium, **4** University Medicine Greifswald, Greifswald, Germany, **5** University Greifswald, Greifswald, Germany

☯ These authors contributed equally to this work.
* joachim.hasebrook@steinbeis.de

**Data Availability Statement:** The full results including statistical data output, all surveys and tools as well the final report of the project, are open

## Abstract

### Background

Tele-emergency physicians (TEPs) take an increasingly important role in the need-oriented provision of emergency patient care. To improve emergency medicine in rural areas, we set up the project 'Rural|Rescue', which uses TEPs to restructure professional rescue services using information and communication technologies (ICTs) in order to reduce the therapy-free interval. Successful implementation of ICTs relies on user acceptance and knowledge sharing behavior.

### Method

We conducted a factorial design with active knowledge transfer and technology acceptance as a function of work satisfaction (high vs. low), workload (high vs. low) and point in time (prior to vs. after digitalization). Data were collected via machine readable questionnaires issued to 755 persons (411 pre, 344 post), of which 304 or 40.3% of these persons responded (194 pre, 115 post).

### Results

Technology acceptance was higher after the implementation of TEP for nurses but not for other professions, and it was higher when the workload was high. Regarding active communication and knowledge sharing, employees with low work satisfaction are more likely to share their digital knowledge as compared to employees with high work satisfaction. This is an effect of previous knowledge concerning digitalization: After implementing the new technology, work satisfaction increased for the more experienced employees, but not for the less experienced ones.

### Conclusion

Our research illustrates that employees' workload has an impact on the intention of using digital applications. The higher the workload, the more people are willing to use TEPs.

to the public at: https://innovationsfonds.g-ba.de/beschluesse/landrettung-zukunftsfeste-notfallmedizinische-neuausrichtung-eines-landkreises.32 The complete data set (i.e., the original, raw data) has been anonymized and uploaded as supplementary file.

**Funding:** All authors participated in the funded project. This research has been funded by a €5.38 million grant from Germany's Innovation Fund (Innovationsfonds des Gemeinsamen Bundesausschusses, G-BA) under number 01NVF16004. URL: https://innovationsfonds.g-ba.de/projekte/neue-versorgungsformen/landrettung-zukunftsfeste-notfallmedizinische-neuausrichtung-eines-landkreis.63 The funding institution had no role in study design, data collection and analysis, decision to publish, or preparation of the manuscript.

**Competing interests:** The authors have declared that no competing interests exist.

Regarding active knowledge sharing, we see that employees with low work satisfaction are more likely to share their digital knowledge compared to employees with high work satisfaction. This might be attributed to the Dunning-Kruger effect. Highly knowledgeable employees initially feel uncertain about the change, which translates into temporarily lower work satisfaction. They feel the urge to fill even small knowledge gaps, which in return leads to higher work satisfaction. Those responsible need to acknowledge that digital change affects their employees' workflow and work satisfaction. During such times, employees need time and support to gather information and knowledge in order to cope with digitally changed tasks.

## Introduction

Despite the increasing need for and continuous development of innovations in health care, conceptual approaches are still limited [1, 2] and scientific frameworks are only beginning to evolve [3, 4]. One of the major limiting factors for more innovation in healthcare is not scarcity of innovations but poor innovation dissemination [5, 6]. As existing frameworks for healthcare transition research are mostly domain [4, 7] or country specific [8] or both, a consistent and widely accepted scientific foundation is still lacking [9]. However, previous research has shown that different perspectives of professional groups and roles have to be taken into account [10] as well as the diverging positions of organizations [6]. Moreover, several research studies have identified factors supporting employees to accept and carry digital transformation. This is crucial in patient-related processes in order to ensure continuous high-quality provision and avoid errors or insufficient workflows. Among these factors are high technology acceptance and willingness to share knowledge [11, 12], technology acceptance and job satisfaction [13, 14], job satisfaction and knowledge sharing [15, 16] and a combination of job satisfaction and dissatisfaction with technology acceptance and knowledge sharing [17–19]. Although it seems evident that all these factors may play an important role for different professional groups involved in a digital transformation process, it is hardly possible to systematically observe and evaluate the interplay of these factors in a relevant medical setting.

### Rural|Rescue: A project to restructure rural rescue services

German emergency medical services (EMS) are a two-tiered system with paramedic-staffed ambulances as primary response supported by pre-hospital emergency doctors for life threatening conditions. Medical practitioners are in short supply whilst the demand for timely emergency medical care is constantly growing. In rural areas this has led to critical delays in the provision of emergency medical care. In rural areas in Germany, the statutory EMS response time (EMS-RT, time interval from the first call to arrival of the EMS) is ten minutes for ambulances and fifteen minutes for pre-hospital emergency doctors respectively. Analyses for rural districts have shown that in many regions the statutory EMS-RT are not met. In particular, in cases of cardiac arrest, time is of the essence because, with each passing minute, the chance of survival with good neurological outcome decreases [20].

The digital transformation introduced to the emergency patient care sector in the county of Vorpommern-Greifswald in North-Eastern Germany provided an exceptional opportunity to evaluate success factors, such as job (dis-)satisfaction, knowledge sharing and technology acceptance among different professional groups. The project Land|Rettung (English: Rural|Rescue) restructured professional rescue services using information and communication

technologies (ICTs) in order to reduce the therapy-free interval. The project is among the first projects to realign emergency medicine in rural areas applying telemedicine and the first large scale transformation project in Germany. As one part of this restructuring, tele-emergency physicians (TEPs) were introduced. The TEP concept increases the quality of patient care and offers the possibility to overcome medical provision problems that arise because of the shortage of emergency physicians [21–24]. TEPs are specially trained emergency physicians whom paramedics and other emergency physicians can contact in order to ask for support. From their desks in a central office, TEPs can guide and supervise the personnel at the emergency site, as they receive the patients' vital signs in real time. Additionally, a video can be transmitted. TEPs can authorize and supervise the administration of drugs and medical interventions [25]. This TEP concept ensures high-quality treatment by emergency physicians, even if they are not on site. Furthermore, the concept is able to reduce the response time between emergency alert and emergency vehicle on scene as well as the critical "therapy-free-interval" [26].

The employees' commitment to work with mostly unknown TEPs enables the project's success. To ensure a successful implementation of such a concept within an existing workflow, it is crucial for organizational leaders to ascertain that the involved groups accept and support the digital transformation. Employees experience insecurities in the wake of digital changes, as they rapidly need to adapt to (drastically) changing processes and workflows. Therefore, employee acceptance can be seen in two different kinds of actions: 1) developing the intent to use the new digital instrument (passive) and 2) starting to share the growing knowledge about the instrument in order to actively spread experiences made in early stages of the implementation.

## Theoretical background and hypothesis development

**Satisfaction, workload, and the intention to use new technologies.**   Employee satisfaction and work output are directly related, as shown by Hackman & Oldham's Job Characteristics Model [27] and Job Demand-Control Model by Karasek [28]: Satisfaction increases with variety of required skills, importance of work for the professional and social environment, autonomy in the execution of tasks and feedback on results. Stress levels rise when work requirements are perceived as high and autonomy in the workplace is perceived as low. Factors that allow for more controllability, decision-making and freedom of the actual work design lead to stress reduction. It is important that job requirements and work resources are balanced [29].

The employees' intention to use new instruments depends on several factors such as the performance expectancy the individual has towards the new technology (perceived usefulness) or the effort employees expect to invest in order to be able to use the new application (perceived ease of use) [30]. Besides those expectations, stress-related factors can also have a significant effect on the employees' intention to use new applications. Stress may stem from two different origins: 1) anxiety, meaning low technology-related self-efficacy or 2) high workload [31]. IT anxiousness describes employees that are uneasy or afraid to use new digital technologies and applications [32]. This leads to stress for the employee at the workplace. Naturally, anxiety hinders employees to participate, which significantly lowers the intention to use the technology [33].

A high workload leads to a high amount of stress. Various studies have found that this workload-related stress is a factor influencing the intention to use new technologies or instruments [31, 34, 35]. Hsiao and Chen [35] describe that it is necessary for medical organizations to reduce physicians' workload in order to increase the physicians' intention to use computerized clinical practice guidelines. Here, workload and stress are negatively correlated with the

willingness to use the new application. El Halabieh, Beaudry, and Tamblyn [31] found similar effects examining the implementation of a digital drug management system. Accordingly, high workload and stress are risk factors in the implementation process of new digital solutions.

**Importance of communication and knowledge sharing.** While the individual intention to use new technologies is a crucial step towards the successful implementation of a new application, the knowledge sharing process among coworkers may also be important. Knowledge is an asset for organizations, especially nowadays. In order to be successful, companies need to assure that useful knowledge is available to all employees and decision-makers across the organization [36]. Knowledge is mostly shared at the individual level [37] within or across teams and departments. As a powerful resource, employees can utilize knowledge strategically. They can decide whether they want to share their expertise, which consequently benefits the whole organization [36], or keep the knowledge to themselves in order to appear irreplaceable [38].

Previous research in various fields refers to aspects influencing the probability of knowledge sharing activities in organizations. One prominent factor affecting knowledge sharing is the employees' overall motivation and encouragement [39]. In general, employees' motivation to engage in knowledge sharing activities seems to determine success [40]. It can be generated through external incentives and rewards, as knowledge sharing requires significant efforts for employees [38]. In order to increase the probability of sharing, organizations should compensate those efforts and costs [41]. Mullins [42] emphasizes that employees appreciate monetary rewards. On the other hand, receiving recognition from the employer or other intrinsic rewards drives optimizing performance and efforts towards the organizations' goal [43]. Being recognized and experiencing that one's knowledge is beneficial to company's progress leads to employees developing a higher interest in sharing their expertise [40, 44].

This perception is characteristic of high self-efficacy [45]. Bandura [46] defines self-efficacy as "the belief in one's capabilities to organize and execute the courses of action required to manage prospective situations." Believing in one's abilities to perform well and reaching certain objectives [47] must be supported and promoted by the employer [39]. This is fundamental in order to achieve successful knowledge sharing. Low self-efficacy–for example because of discrimination–leads to less knowledge sharing [48].

However, it appears that not only monetary or intrinsic motivation is a factor influencing knowledge sharing. Knowledge sharing behavior is highly dependent on the social environment it is practiced in [48]. They found that not sharing the same social identity leads to negative knowledge sharing behavior. Identification which is defined as "the process whereby individuals see themselves as grouped with another person or set of people" [41] appears to be a highly influential aspect regarding knowledge sharing. The more people can identify with one another, the more information they share [15, 45, 49]. It is therefore crucial that employees build relationships with colleagues, further develop strong connections and establish a comfortable basis to share thoughts and expertise [40, 41]. Organizations can promote this basis for intense and frequent knowledge exchange. They can show social recognition [50] and facilitate frequently low-threshold communication [51]. Although various researchers found a link between social networks and knowledge sharing behavior, other research contradicts these findings. Wilkesmann, Wilkesmann and Virgililito [38] for example state that team orientation has no effect on the probability of knowledge sharing. Furthermore, social pressure does not appear to influence knowledge sharing [40]. Regarding the influence of trust in colleagues and the organization on knowledge sharing activities, research delivers contradicting findings. Whereas a few papers did not find a positive effect of trust on employees' knowledge sharing behavior [40, 41], others were able to show that trust has a positive effect [45]. Huemer, von Krogh and Roos [52] emphasized that trust among coworkers facilitates learning within the organization and sharing knowledge earlier.

**Research model and hypotheses.** As mentioned above, high technology acceptance, knowledge sharing, job satisfaction or dissatisfaction, stress and workload are all relevant factors to ensure high quality health care and avoid insufficiency or even errors during a technical transformation process [13–19]. Hsiao and Chen [31] are pointing out that there are three levels influencing healthcare workers' intention to use computerized clinical practices: The technology itself, subjective and environmental factors. The Technology Acceptance Model (TAM) provided as a conceptual model to measure the intention to use and actual use of digital technologies [53]. It is based on the Theory of Reasoned Action (TRA) explaining an attitude toward a certain behavior by positive or negative feelings (evaluative effect), and subjective norms describing a person's perception of what he or she is expected to do [54]. It also integrates the theory of planned behavior (TPB), an extension of TRA, adding perceived behavioral control as an important aspect [55]. According to TAM, perceived usefulness, perceived ease of use, and the complexity of the system are relevant features on a technological level. On a subjective level attitude, prior experience, and task uncertainty are important issues. On an environmental or organizational level, social influence or peers and superiors as well as perceived organizational support are critical for the intention to use and actual use of digital technologies.

The subjective and organizational level can be described in terms of the Job Demands-Resources Model (JDR, [56]), which was originally developed to explain burnout but has been applied to a large variety of jobs, skills, and working conditions [57]. According to JDR, work factors can be distinguished into job demands and job resources [58]. Job demands are associated with physical and/or psychological effort, especially workload [59] or a complex and unfavorable physical or technical environment [57]. Job resources are physical, psychological, social, or organizational aspects helping to achieve work goals or reduce job demands, especially perceived social or organizational support [57]. Sufficient resources meeting low demands lead to high satisfaction and low perceived workload resulting in more collaboration and knowledge sharing [39] and easier acceptance of change and new technologies. However, if insufficient resources are available to satisfy high demands, this leads to stress and a perceived high workload resulting in less knowledge sharing and acceptance of demanding new technologies [56, 60].

Considering the points made in previous research, we examine the impact of employees' perception of stress and workload on the intention to use TEPs.

Hypothesis 1: The more stressed the employees are, the less open-minded they are towards technological change [33, 57, 60].

Secondly, we examine whether employees' job satisfaction correlates with knowledge sharing activities. Hypothesis 2: The more the employee trusts his or her employer and is internally motivated, the more he or she is likely to share valuable knowledge [39, 56, 59].

We assume, that this also account for situations, in which rapid changes and breaks in the organization occur. For that reason, the third hypothesis reads as follows:

Hypothesis 3: Employees with higher job satisfaction are more likely to communicate and to share their knowledge with others in order to overcome temporary, organization-wide task uncertainty than people with lower job satisfaction [17–19, 53].

## Method

### Design

We examined the impact of work satisfaction and workload on active knowledge transfer and technology acceptance in a pre-post design prior to and after digital transformation of emergency care resulting in a 2x2x2 design: active communication and knowledge sharing as well

as technology acceptance as a function of (i) work satisfaction (high vs. low), (ii) workload (high vs. low) and (iii) point in time (pre vs. post). Material and procedure of this study were approved by the Ethics Commission of the University Medicine Greifswald (approval BB 044/17). All subjects were informed about their rights and signed consent forms, which were collected and archived during the project duration (S1 File).

## Subjects

To assess technology acceptance, employees of all organizations involved or affected by the implementation of TEPs in the study region were approached. This included paramedics of the six ambulances equipped with tele-emergency technology, collaborating paramedics and emergency physicians, emergency dispatchers and nurses and doctors at the emergency departments of local hospitals. There are three different paramedic qualification levels in Germany: basic, intermediate and advanced. The advanced level of paramedics was established in 2013 and since then a high number of intermediate paramedics qualified further to become advanced paramedics [61]. In order to enable a before-and-after comparison, the survey was done before the introduction of the tele-emergency medical system and repeated two years later. During the first phase of the study (pre), 411 surveys were issued as machine readable questionnaires (S2 File) over a period of three months, of which 194 completed and evaluable questionnaires were returned (response rate 47.2%). Due to shift work, vacation and fluctuation rates up to 25%, a response rates close 50% is comparably high and might document the high interest the paramedics took in the project [62]. As the number of 411 surveys reflects almost the entire relevant population of paramedics of the relevant ambulances, the participation of 79 subjects would have been sufficient to reach an acceptable confidence level of 5% and test power of 90%. The response rate of 47.2% (194 subjects) in the pre-part of the study results in an estimated marginal error of 5.3%.

The second measurement (post) took place two years after the finalized introduction of the tele-emergency system. The population of respondents decreased between pre and post surveys by about 20%. This is due to a restructuring both on the guards and in the emergency room of a medical center, where a relatively high number of freelancers was replaced by a smaller number of permanent staff. During the second phase of the study (post), 344 employees were approached, and 115 evaluable questionnaires were returned (response rate 33.4%). In the post-phase, 76 subjects would suffice to ensure a confidence level of 5% and a test power of at least 90%. The response rate of 33.4% (115 subjects) results in a marginal error of 7.5%. A response rate of over 30% is lower than in the first phase but still satisfactory as compared to response rates in other studies [62]. There is no indication for any non-responder biases. The lower response rate seems to reflect the higher workload of the permanent staff and a decreased interest in the study about the TEP system, which had been in place more than one and a half year, at that point of time.

All subjects were asked to indicate the type of organization of their employer, their profession, and their gender (see Table 1). Due to data protection issues no other specifications were gathered.

## Material

**Job satisfaction and job demands.** In an interview study with 178 interviews about retention and intention-to-leave, we identified central aspects of satisfaction and dissatisfaction of professional medical staff [63–66] and combined these aspects in a polarity profile [64, 67], which includes questions to be scored from 1 (positive pole) to 10 (negative pole; see Table 2). An example question concerning support from colleagues reads as follows: "For my work I get

**Table 1. Participants' characteristics in the pre and the post phase.**

|  | Pre | | Post | |
| --- | --- | --- | --- | --- |
|  | **n** | **%** | **n** | **%** |
| *Employer / type of organization* | | | | |
| Total | 194 | 100 | 115 | 100 |
| medical dispatch center | 23 | 11.9 | 17 | 14.8 |
| medical emergency service | 123 | 63.4 | 71 | 61.7 |
| emergency department | 38 | 19.6 | 19 | 16.5 |
| administration | 2 | 1.0 | 1 | 0.9 |
| other | 6 | 3.1 | 2 | 1.7 |
| not specified | 2 | 1.0 | 5 | 4.3 |
| *Profession* | | | | |
| Total | 194 | 100 | 115 | 100 |
| Physicians | 57 | 29.4 | 24 | 20.9 |
| Nurses | 23 | 11.9 | 17 | 14.8 |
| paramedic (advanced) | 21 | 10.8 | 5 | 4.3 |
| paramedic (intermediate) | 47 | 24.2 | 28 | 24.3 |
| paramedic (basic) | 14 | 7.2 | 21 | 18.3 |
| emergency dispatcher | 21 | 10.8 | 15 | 13.0 |
| other | 8 | 4.1 | 4 | 3.5 |
| not specified | 3 | 1.5 | 1 | 0.9 |
| *Gender* | | | | |
| total | 194 | 100 | 115 | 100 |
| male | 129 | 66.5 | 72 | 62.6 |
| female | 55 | 28.4 | 26 | 22.6 |
| not specified | 10 | 5.2 | 17 | 14.7 |

sufficient appreciation and support from my colleagues" (positive pole = 1) or "My work is not appreciated and unnecessarily criticized by my colleagues" (negative pole = 10). The scale is highly reliable with Cronbach's Alpha = 0.91 in this and 0.78 a prior study [66, 67]. Discriminant analyses show a perfect match of the scale with intention to stay as well as promotion scores of working conditions (100% correct classification) supporting its convergent validity. The different aspects reflected in the scale show only little overlap indicated by significant intercorrelations. Also, high satisfaction was correlated with high frequencies of positive statements and dissatisfaction with high frequencies of negative statements suggesting a good discriminant validity [66, 67].

**Technology acceptance.** The increasingly important acceptance of technology in the course of digitalization of medical care has been examined with different versions of the Technology Acceptance Model (TAM). The concept was originally proposed by Davis in his thesis [68] and several times extended by Venkatesh and Davis [69, 70] and now integrates a great number of relevant factors [71]. It has been argued that the original TAM was too simple to describe relevant aspects of technology acceptance whereas extended versions of TAM and UTAUT are too complex [72] and neglect the medical context in which technology is used [73]. Therefore, we focus on the main aspects determining the intention to use a system, that is, perceived ease and perceived usefulness of the system and adapted items from an extended TAM (TAM2, [69] to the specific context of the TEP system [see 74; Table 3, 75]), with statements like: "I think that the concept of the tele-emergency doctor leads to relevant time-savings" for ease of use or: "I think the concept of the tele-emergency doctor improves my professional performance" to determine the usefulness (see Table 6 for a full list of all 19

**Table 2. Polarity profile for job satisfaction and workload.**

| Positive Pole | Negative Pole |
|---|---|
| *Job Satisfaction* | |
| For my work I get sufficient appreciation and support from my colleagues. | My work is not appreciated by my colleagues and unnecessarily criticized. |
| For my work I get sufficient appreciation and support from my superiors. | My work is not appreciated by my superiors and unnecessarily criticized. |
| I will be informed in timely and sufficient manner about plans and decisions concerning my work. | I am not informed in a timely and sufficient manner about important plans and decisions. |
| Decisions concerning my work as well as the decision-making process are easy for me to understand. | I cannot understand decisions and decision-making processes that affect me. |
| The work offers many challenges, but I never feel overwhelmed. | I feel overwhelmed by the demands of my work. |
| As part of my employment in emergency care, I am doing a meaningful job that benefits society. | I do pointless work that doesn't benefit anyone. |
| Only with my current employer do I find the working conditions that are important to me and which I want to have. | I might as well work for another employer in emergency care. |
| For my work in the context of emergency care I am paid fairly and appropriately. | I am not sufficiently paid for the work I do in the context of emergency care. |
| My current position in emergency care offers me optimal opportunities to develop and pursue a career in my profession. | I see my current position in emergency care as a dead end in which I cannot develop professionally. |
| *additional item "workload"* | |
| In my experience, the workload at my employer in emergency care is not too high and will continue to be the case. | In my experience, the workload is unbearable, and it will not improve in the future. |

**Table 3. Scale on active individual knowledge transfer based on expert ratings (with average rating and standard deviation).**

| | Item | Average rating | Std. dev. |
|---|---|---|---|
| 0 | The training in the use of new work equipment, new software, etc. usually takes place through personal exchange and observation at the workplace. *(removed item)* | 4.20 | 0.92 |
| 1 | If you can explain something badly with words, but have to present e. g. operation of new devices, there is always a colleague who takes over. | 4.00 | 1.20 |
| 2 | In our organization, cross-team or cross-group meetings are held regularly in order to improve cooperation among each other. | 4.30 | 0.95 |
| 3 | When making important decisions, each employee can make his or her own arguments and is involved in the decision-making process | 4.20 | 0.42 |
| 4 | In our organization, even short-term teams and shift groups coordinate quickly and then work together productively and smoothly. | 4.70 | 0.67 |
| 5 | There are internal quality circles on internal innovations and idea management, which serve in our organization for continuous improvement and quality assurance. | 4.20 | 0.63 |
| 6 | New employees receive advice and induction support for their job in our organization. | 4.50 | 0.71 |
| 7 | There are sufficient offers for the exchange of knowledge between organizations that have common interests (e. g. interdisciplinary training, case conferences). | 4.10 | 0.88 |
| 8 | Our organization offers internal seminars, workshops, etc., which are led by internal experts of the company. | 4.40 | 0.52 |
| 9 | If I would like to contribute to an offer of information myself, I believe that this is possible in an uncomplicated and fast way, as there are sufficient possibilities. | 4.20 | 1.03 |
| 10 | From my point of view, participation in information supplies, i.e., accession, co-authoring, etc., always happens uncomplicatedly and without much delay. *(added item)* | 3.80 | 1.32 |
| 11 | Participation in information sessions significantly improves my skills for solving problems and tasks in everyday work. *(added item)* | 3.80 | 1.43 |

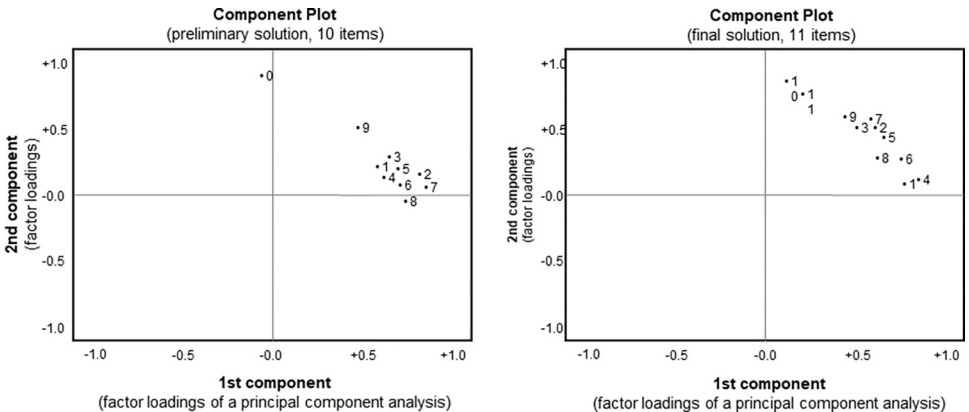

**Fig 1. Factor analyses to identify the items of the scale on active individual knowledge transfer.** left: preliminary solution, right: final solution; numbers refer to item numbers in Table 3.

items). The participants responded to them on a Likert scale from consent (1 = fully agree) to rejection (4 = do not agree at all). The scale shows a highly reliability (Cronbach alpha = 0.87) in this study. In other studies, the TAM2 scale showed high divergent and convergent validity (standardized loading >0.5 and composite reliability >1.96; [76]).

**Measuring knowledge sharing.**   Following Hardwig [77, 78], we distinguished between knowledge types, knowledge sources and knowledge instruments. Based on Erpenbeck [79], the transfer of implicit and explicit knowledge was examined at the individual level (from one person to another person) and at organizational level (e.g. from department to department). The types of knowledge were categorized according to the KODE Competence Atlas [80] comprising methodological, social and personal competences. In the case of knowledge sources, a distinction was made between internal and external sources and whether those were linked to personal, organizational or documentary sources. The evaluation was carried out on a 6-step Likert scale from consent (1 = fully agree) to rejection (6 = do not agree at all; full list of all 46 items in S1 Appendix).

The appropriateness of these items to indicate active individual knowledge transfer, was rated by eleven subject matter experts from Steinbeis University and Curacon, who were not involved in the Rural|Rescue project, (from 1 = fully disagree to 5 = fully agree). High consensus was reached with an average intraclass correlation of 0.92 (F[7,210] = 18.4; p < .001). While selecting all items with an average value higher than 4 a factor analysis revealed that the new scale was not consistent due to an outlying item (cf. Fig 1, left hand side). Removing this item and replacing it by two items with an average rating higher than 3.8 resulted in a highly consistent scale (Fig 1, right hand side; Cronbach alpha = 0.76) on active individual knowledge transfer comprising 11 items (see Table 5; the full factor structure is documented in S2 Appendix). High consensus of the raters, and the selection of items with a high rating (>3.8) supports the scale's content validity [81], the uniform factor structure its construct validity [82]. The selective impact of satisfaction, professional role, and prior experience on knowledge sharing reported later in this study supports the criterion validity of the scale [83], which was selected from a comprehensive research instrument on various aspects of knowledge sharing [77, 78].

## Procedure

A board variety of organizations and institutions have been involved in the RuralRescue project: State and district authorities, governmental associations, such as the medical chamber,

non-governmental organizations, such as German Red Cross, state-owned and municipal as well as privately owned hospitals. The project was directed by the head of the district authorities (Landrat), who informed all affected organizations and stakeholders. They were informed about purpose and timeline of the project in writing and by a series of conferences and regional workshops. All institutions, which were asked to take part in the study, received preliminary versions of the survey and gave feedback, which was incorporated in the final version of the survey. The final version of the survey was printed as machine-readable questionnaires, which were carried and presented to the subjects by a member of the project team. He or she handed out the consent form and the surveys and collected the consent forms. The surveys were put in a sealed enveloped directed to the project team by the subjects themselves.

Prior to the introduction of the telemedical system, questionnaires were issued over a period of three months to subjects employed in the rescue chain including paramedics, physicians, dispatchers, and nurses. The second measurement took place 24 months later over a period of three months after completely introducing the tele-emergency system. Between the first and the second measurement, physicians, nurses and paramedics as well as dispatchers and administrative staff were trained to use the TEP system based on experiences in prior projects [84] and standards defined in these projects [85]. Only physicians with at least 500 emergency responses and additional qualifications according to the European Resuscitation Council (ERC), Pre-Hospital Trauma Life Support (PHTLS), or Advanced Trauma Life Support (ATLS) qualified to serve as TEP.

Sufficient training is essential for the successful introduction of telemedicine in emergency care. Therefore, the training consists of several layers: The initial training consists of 5 days of demonstrations and simulations under the supervision of an experienced TEP. Further updates and trainings were provided via simulation training and e-learning modules throughout the entire project period. In addition, some experienced TEPs were trained as TEP supervisors supporting their TEP colleagues as well as paramedics working with the TEP system. These supervisors educated experienced paramedics as disseminators and promoters, who introduced their colleagues to the TEP system in one-day courses and trained them in the use of the system via live demonstrations and simulation exercises. A detailed description of the implementation and training phase can be found in [86].

## Results

### Job satisfaction, workload, and professional role

The dependent variables of job satisfaction (from 1 = positive to 10 = negative pole), workload (from 1 = positive to 10 = negative pole), technology acceptance (from 1 = full acceptance to 4 = low or no acceptance) and knowledge transfer (from 1 = low or no transfer to 5 = high transfer) were calculated as a function of the professional role (Table 4). A general linear model (GLM) with the professional roles as independent variables resulted in three main effects: job satisfaction ($F[7,293] = 3.29$; $p < .01$), technology acceptance ($F[7,293] = 4.81$; $p < .001$) and active knowledge transfer ($F[7,293] = 6.99$; $p < .001$) indicating higher job satisfaction and technology acceptance for paramedics than for physicians and nurses and the highest active knowledge transfer amongst physicians. There was an interaction with technology acceptance and point in time (pre vs. post; $F[7,286] = 5.82$; $p < .001$) due to a significant improvement of technology acceptance for nurses (2.60 vs. 3.05; $p < .001$ Scheffé) but not for other professions.

### Technology acceptance

The independent variables "Satisfaction" and "Workload" were split up based on the median in "low" (median or lower) and "high" (higher than the median). A GLM was calculated with

**Table 4. Means (and standard deviations) of job satisfaction, workload, technology acceptance and active knowledge transfer as a function of professional role.**

| | Job satisfaction | Workload | Knowledge transfer | Technology acceptance | |
|---|---|---|---|---|---|
| | | | | pre | post |
| Physicians | 4.20 (1.69) | 4.68 (2.44) | 3.30 (0.77) | 2.64 (0.47) | 2.24 (0.41) |
| Nurses | 4.39 (1.32) | 5.76 (2.32) | 2.93 (0.61) | 2.60 (0.32) | 3.05 (0.34) |
| paramedic (advanced) | 3.85 (1.52) | 4.43 (2.66) | 3.23 (0.83) | 2.23 (0.54) | 2.17 (0.48) |
| paramedic (intermediate) | 4.25 (1.52) | 4.52 (2.19) | 3.43 (0.84) | 2.45 (0.57) | 2.40 (0.40) |
| paramedic (basic) | 4.64 (1.53) | 4.82 (2.55) | 3.34 (0.96) | 2.45 (0.55) | 2.38 (0.48) |
| emergency dispatcher | 3.43 (1.43) | 4.25 (2.21) | 2.44 (0.69) | 2.40 (0.37) | 2.30 (0.47) |

"Time", "Satisfaction" and "Workload" as independent and the mean of the scales "Technology acceptance" and "Active knowledge transfer" as dependent variables. We describe the results in two separate sections, first on technology acceptance followed by knowledge transfer. Results are reported on item level followed by the means relevant for the significant effects. A complete breakdown of all means can be found in S3 Appendix.

Only one significant main effect was found: technology acceptance was generally on a medium level (2.45 overall from 1 best to 4 worst score), and it was better when the workload was high (2.66 vs. 2.34; F[1,321] = 11.33; p < .001; Table 5, bottom, right hand side). An inspection of the differences between all single items shows, that this somewhat surprising effect is based on a generally more positive evaluation of the colleagues' opinion (item 3), patient care (item 14), performance (item 15) and usefulness in general (item 17)–although persons with a high workload also think that the TEP concept interferes with established organizational structures (item 18; all differences significant after Bonferroni-Holmes correction).

## Active knowledge transfer

The same GLM has been calculated with "Time", "Satisfaction" and "Workload" as independent and "Technology acceptance" and "Active knowledge transfer" as dependent variables revealed significant effects. Active knowledge transfer resulted only in one significant main effect: knowledge transfer was higher when satisfaction was lower (F[1,309] = 26.57; p < .001). This highly significant effect clearly contradicts our expectations (higher satisfaction leads to more knowledge transfer).

In order to further investigate the contradictory effect of job satisfaction, we split up the factor "Satisfaction" into four quartile groups instead of only two groups below (low) and above (high) the median (cf. Fig 2). This analysis showed an almost linear decrease of knowledge transfer rates as a function of satisfaction (from 2.64 transfer rate for highest satisfaction to 3.72 for lowest satisfaction; F[3,320] = 27.39; p < .001).

We further investigated the impact of the factors „point in time"(pre vs. post) and „leading position"(leading vs. non leading) related to satisfaction and knowledge transfer. No factor caused an effect on knowledge transfer, but two interactions were revealed: First, point in time (pre vs. post) and leading position (leading vs. non leading) showed a significant interaction (F[1,174] = 4.87; p < .05; see Fig 3 and Table 6).

An investigation of the differences on item level shows that almost all items concerning knowledge transfer differ for persons with a high and low job satisfaction, respectively–but only when they are I a non-leading position (Table 6, bottom right section; all differences significant after Bonferroni-Holmes correction).

Again, we split up the satisfaction in quartile groups and examined the same interaction effect: A significant three-way interaction (F[1,174] = 5.27; p < .05) was found between point in time (pre vs. post), position (leading vs. non leading) and the four quartile groups of

**Table 5. Items concerning acceptance of the TEP system with means (and standard deviations) as a function of time (pre, post) and workload (low, high).**

| Items | Time | | | Workload | | |
|---|---|---|---|---|---|---|
| | Pre | Post | sig.diff.* | Low | High | sig.diff.* |
| 1. I think the concept of the tele-emergency physician (TEP) makes sense. | 2.05 (0.88) | 1.79 (.91) | n.s. | 2.09 (0.94) | 1.90 (0.87) | n.s. |
| 2. Superiors to me consider the concept of the tele-emergency doctor to be useful. | 1.87 (0.78) | 1.86 (.78) | n.s. | 1.93 (0.79) | 1.83 (0.77) | n.s. |
| 3. Colleagues, who are important to me, consider the concept of the TEP to be useful. | 2.28 (0.84) | 2.02 (.85) | n.s. | 2.38 (0.87) | 2.09 (0.83) | p < .05 |
| 4. People in my circle of friends think the concept of the TEP makes sense. | 2.57 (0.95) | 2.18 (.96) | p < .001 | 2.48 (0.94) | 2.39 (0.98) | n.s. |
| 5. When I participate in a project such as the TEP, my reputation among colleagues increases. | 3.12 (0.89) | 3.15 (.87) | n.s. | 3.30 (0.90) | 3.04 (0.85) | n.s. |
| 6. When I take part in a project such as the TEP, my reputation among friends increases. | 3.15 (0.91) | 3.08 (.97) | n.s. | 3.33 (0.89) | 3.03 (0.93) | n.s. |
| 7. I think that the concept of the TEP leads to a relevant time saving. | 2.36 (0.94) | 2.25 (1.09) | n.s. | 2.51 (0.98) | 2.22 (0.99) | n.s. |
| 8. I think that the concept of the TEP leads to a faster diagnosis finding. | 2.27 (0.89) | 2.27 (0.99) | n.s. | 2.45 (0.97) | 2.18 (0.89) | n.s. |
| 9. I think that the concept of the TEP leads to a faster start of therapy. | 2.07 (0.85) | 1.97 (0.94) | n.s. | 2.21 (0.92) | 1.95 (0.86) | n.s. |
| 10. I think that the concept of the TEP leads to a faster transportability. | 2.35 (0.94) | 2.31 (0.97) | n.s. | 2.43 (0.99) | 2.29 (0.93) | n.s. |
| 11. I think that the concept of the TEP reduces my documentation effort. | 3.16 (0.82) | 2.55 (1.14) | p < .001 | 3.04 (0.97) | 2.89 (0.99) | n.s. |
| 12. I think that the concept of the TEP reduces my workload. | 3.05 (0.86) | 2.76 (0.99) | p < .05 | 3.10 (0.89) | 2.87 (0.92) | n.s. |
| 13. I think that the concept of the TEP leads to a delay at the emergency site. | 2.69 (0.86) | 2.65 (0.95) | n.s. | 2.63 (0.94) | 2.70 (0.87) | n.s. |
| 14. I think that the concept of the TEP improves the quality of patient care. | 2.15 (0.75) | 1.93 (0.85) | n.s. | 2.28 (0.80) | 1.97 (0.77) | p < .001 |
| 15. I think that the concept of the TEP improves my professional performance. | 2.75 (0.94) | 2.59 (1.08) | n.s. | 2.92 (0.98) | 2.57 (0.98) | p < .01 |
| 16. I think that the concept of the TEP increases my effectiveness at work. | 2.61 (0.93) | 2.53 (1.03) | n.s. | 2.71 (0.95) | 2.51 (0.96) | n.s. |
| 17. I think that the concept of the TEP is useful for my work. | 2.30 (0.86) | 2.08 (0.96) | n.s. | 2.45 (0.92) | 2.11 (0.87) | p < .001 |
| 18. I think that the concept of the TEP interferes with the established structure. | 2.72 (0.81) | 2.99 (0.96) | n.s. | 2.63 (0.88) | 2.91 (0.85) | p < .05 |
| 19. I think that the TEP is also alerted in situations where normally an emergency doctor is not alerted. | 2.20 (0.91) | 2.38 (0.95) | n.s. | 2.12 (0.99) | 2.34 (0.88) | n.s. |
| Overall mean | 2.56 (0.80) | 2.26 (.0,82) | n.s. | 2.66 (0.76) | 2.34 (0.83) | p < .001 |

* sig. diff. indicates significance levels (p< = .05 or not significant) of the differences between the factor levels

satisfaction. It showed higher transfer rates for persons in leading positions with low satisfaction rates especially at the start of the project and for dissatisfied non-leading persons–although on a lower level–at the end of the project (see Table 7).

## Impact of experience level

The results show that emergency personnel with relatively low work satisfaction are more likely to share information with other staff members in times of digital transformation

**Table 6. Items concerning knowledge transfer with means (and standard deviations) as a function of position (leading vs. non leading), time (pre, post) and satisfaction (low, high).**

| Items | Time | | | Satisfaction | | |
|---|---|---|---|---|---|---|
| | Pre | Post | sig. diff.* | Low | High | sig. diff.* |
| *Leading position* | | | | | | |
| 1. If you can explain something badly with words, but have to present e. g. operation of new devices, there is always a colleague who takes over. | 3.67 (1.32) | 1.92 (0.99) | n.s. | 3.33 (1.24) | 3.56 (1.33) | n.s. |
| 2. In our organization, cross-team or cross-group meetings are held regularly in order to improve cooperation among each other. | 3.24 (145) | 1.62 (1.76) | n.s. | 3.00 (1.38) | 3.00 (2.12) | n.s. |
| 3. When making important decisions, each employee can make his or her own arguments and is involved in the decision-making process | 2.95 (1.53) | 2.42 (1.00) | n.s. | 2.96 (1.30) | 3.56 (1.51) | n.s. |
| 4. In our organization, even short-term teams and shift groups coordinate quickly and then work together productively and smoothly. | 3.38 (0.92) | 2.33 (0.65) | n.s. | 3.21 (0.83) | 3.78 (0.67) | n.s. |
| 5. There are internal quality circles on internal innovations and idea management, which serve in our organization for continuous improvement and quality assurance. | 2.90 (1.30) | 1.33 (1.15) | n.s. | 2.83 (1.20) | 2.33 (1.41) | n.s. |
| 6. New employees receive advice and induction support for their job in our organization. | 3.76 (1.09) | 3.00 (0.61) | n.s. | 3.92 (0.88) | 3.67 (1.12) | n.s. |
| 7. There are sufficient offers for the exchange of knowledge between organizations that have common interests (e. g. interdisciplinary training, case conferences). | 2.19 (0.98) | 1.33 (1.44) | n.s. | 2.46 (0.98) | 1.67 (1.32) | n.s. |
| 8. Our organization offers internal seminars, workshops, etc., which are led by internal experts of the company. | 3.10 (1.18) | 1.67 (1.43) | n.s. | 3.13 (1.07) | 2.44 (1.67) | n.s. |
| 9. If I would like to contribute to an offer of information myself, I believe that this is possible in an uncomplicated and fast way, as there are sufficient possibilities. | 3.56 (0.88) | 2.75 (0,45) | n.s. | 3.73 (0.66) | 3.00 (1.41) | n.s. |
| 10. From my point of view, participation in information supplies, i.e., accession, co-authoring, etc., always happens uncomplicatedly and without much delay. (added item) | 3.44 (1.24) | 2.74 (0,49) | n.s. | 3.64 (1.03) | 3.01 (1.40) | n.s. |
| 11. Participation in information sessions significantly improves my skills for solving problems and tasks in everyday work. (added item) | 3.33 (0.87) | 3.00 (0,41) | n.s. | 3.60 (0.89) | 2.99 (1.39) | n.s. |
| Overall mean | 3.19 (0.72) | 2.79 (1.04) | n.s. | 3.03 (0.87) | 3.04 (0.89) | n.s. |
| *Non leading position* | | | | | | |
| 1. If you can explain something badly with words, but have to present e. g. operation of new devices, there is always a colleague who takes over. | 3.58 (1.12) | 2.45 (1.30) | n.s. | 3.78 (1.13) | 3.15 (1.22) | p < .001 |
| 2. In our organization, cross-team or cross-group meetings are held regularly in order to improve cooperation among each other. | 2.26 (1.49) | 1.04 (1.59) | n.s. | 2.50 (1.51) | 1.67 (1.45) | p < .001 |
| 3. When making important decisions, each employee can make his or her own arguments and is involved in the decision-making process | 2.54 (1.46) | 1,48 (1.58) | n.s. | 2.90 (1.37) | 1.98 (1.56) | p < .001 |
| 4. In our organization, even short-term teams and shift groups coordinate quickly and then work together productively and smoothly. | 3.34 (1.28) | 2.38 (1.43) | n.s. | 3.74 (1.14) | 2.89 (1.43) | p < .001 |
| 5. There are internal quality circles on internal innovations and idea management, which serve in our organization for continuous improvement and quality assurance. | 2.59 (1.43) | 1.47 (1.49) | n.s. | 2.77 (1.36) | 2.20 (1.52) | p < .01 |
| 6. New employees receive advice and induction support for their job in our organization. | 3.49 (1.29) | 2.28 (1.43) | n.s. | 3.65 (1.20) | 3.03 (1.48) | p < .001 |
| 7. There are sufficient offers for the exchange of knowledge between organizations that have common interests (e. g. interdisciplinary training, case conferences). | 2.08 (1.28) | 0.98 (1.46) | n.s. | 2.22 (1.31) | 1.75 (1.39) | p < .05 |
| 8. Our organization offers internal seminars, workshops, etc., which are led by internal experts of the company. | 2.75 (1.37) | 1.42 (1.49) | n.s. | 2.82 (1.33) | 2.29 (1.52) | p < .01 |
| 9. If I would like to contribute to an offer of information myself, I believe that this is possible in an uncomplicated and fast way, as there are sufficient possibilities. | 3.38 /1.13) | 2.23 (0.84) | n.s. | 3.39 (0.97) | 3.15 (0.99) | n.s. |
| 10. From my point of view, participation in information supplies, i.e., accession, co-authoring, etc., always happens uncomplicatedly and without much delay. (added item) | 3.09 (1.17) | 2.01 (0.99) | n.s. | 3.11 (1.06) | 2.92 (1.09) | n.s. |
| 11. Participation in information sessions significantly improves my skills for solving problems and tasks in everyday work. (added item) | 3.64 (1.06) | 2.61 (0.65) | n.s. | 3.73 (0.95) | 3.44 (1.19) | n.s. |
| Overall mean | 2.86 (0.88) | 2.71 (0.95) | n.s. | 3.06 (0.74) | 2.41 (1.00) | p < .05 |

* sig. diff. indicates significance levels (p< = .05 or not significant) of the differences between the factor levels

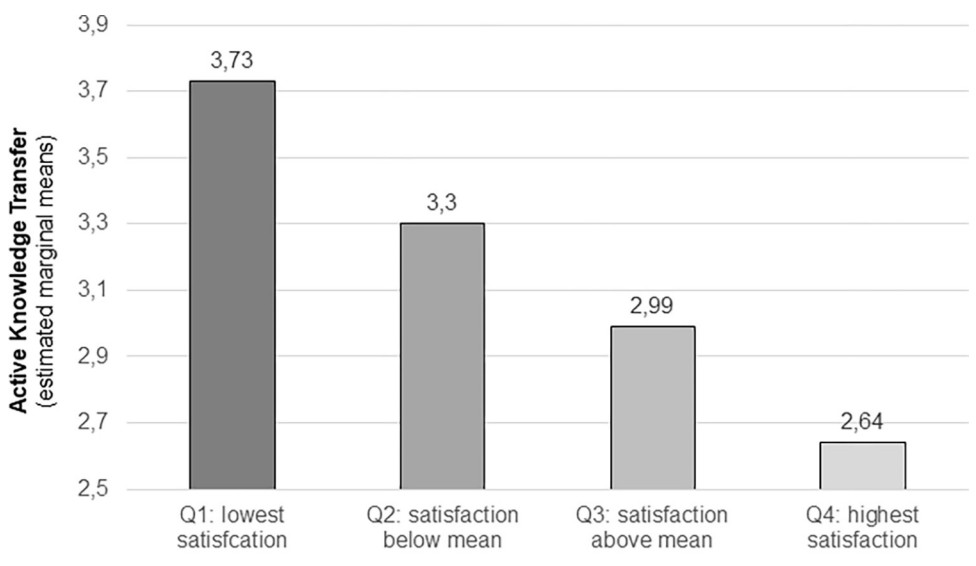

**Fig 2. Effect of satisfaction level (quartile groups from highest to lowest) on active knowledge transfer.**

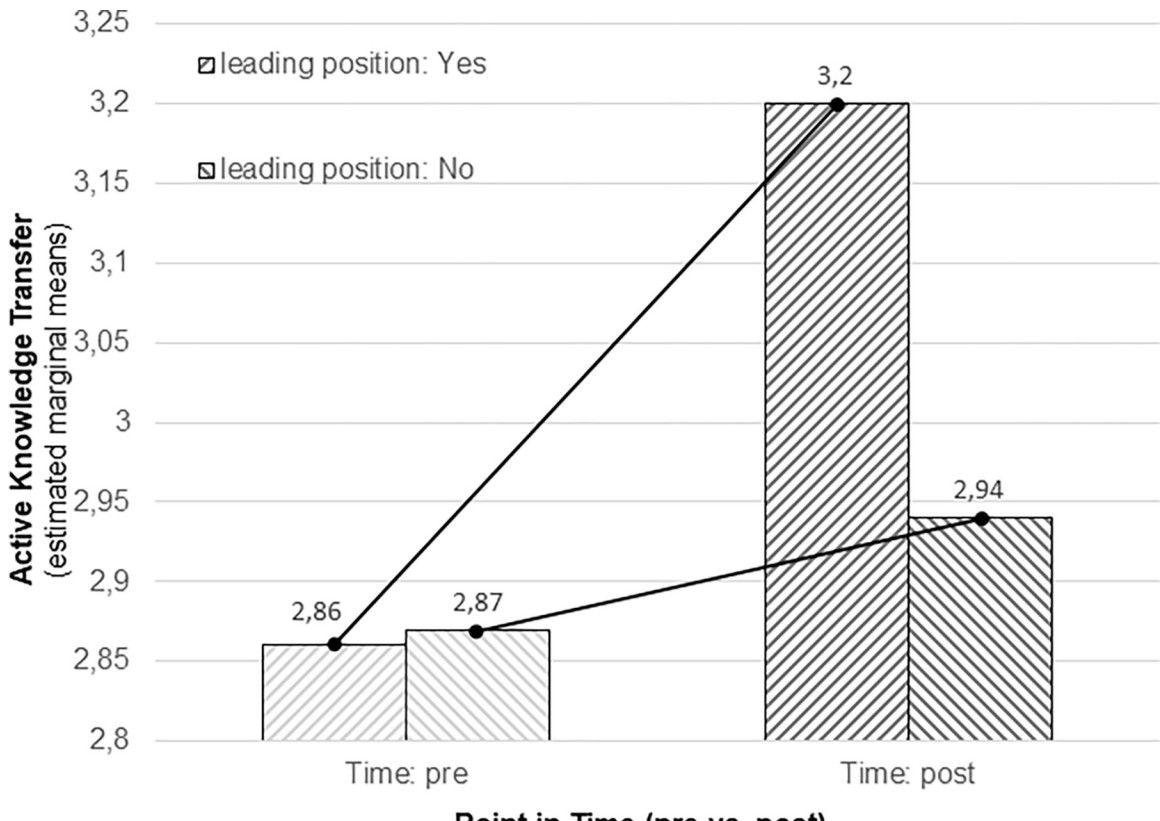

**Fig 3. Active knowledge transfer as a function of point in time (pre vs. post) and management position (leading vs. non leading).**

**Table 7. Means (and standard deviation) for three-way interaction of point in time (pre vs. post), management position (leading vs. non leading) and satisfaction level (quartile 1 to 4) on active knowledge transfer.**

| Point in Time | Leading Position | Satisfaction Level (quartile groups): Mean (Std. Dev.) | | | |
|---|---|---|---|---|---|
| | | 1: lowest | 2: below average | 3: above average | 4: highest |
| Pre | Leading | 4.36 (0.56) | 1.88 (0.73) | 3.31 (0.64) | 2.49 (0.33) |
| | Non leading | 3.54 (0.95) | 3.38 (0.94) | 3.12 (0.78) | 2.67 (0.72) |
| Post | Leading | 3.31 (0.97) | 2.99 (0.55) | 3.89 (1.51) | 2.53 (0.28) |
| | Non leading | 4.02 (1.11) | 3.40 (0.92) | 2.97 (0.76) | 2.89 (0.59) |

regarding their work compared to their more satisfied colleagues. This finding is rather surprising and could be attributed to two different approaches:

1. Staff with low work satisfaction engage in more communication and meta-communication in order to cope with change-related anxiety.

or

2. During organizational change, employees realize their lack of knowledge regarding the new digital system. This leads to lower satisfaction, which employees try to overcome by sharing and building their knowledge further.

If the second assumption is correct, specific knowledge about the TEP solution should counteract the effect of a low satisfaction level on knowledge transfer. We checked the hypothesis using the previously dependent variable "active knowledge transfer" as an independent variable organized in quartile groups and the variable "satisfaction" as a continuous dependent variable. In the survey, we asked whether people had experience with telemedical projects beforehand ("Did you already participate in another telemedical project?"). Employees who had already worked with telemedical solutions were therefore labeled "experienced" which allowed us to check for the impact of experience on satisfaction and knowledge transfer. Consequently, we calculated a GLM with the independent variable knowledge transfer (quartile groups from 1 = low to 4 = high), experience level (high vs. low) and point in time (pre vs. port) and work satisfaction as a dependent variable. The variable „technology acceptance" was taken as a co-variate to control its direct influence on the factors "experience" (yes vs. no) and "time" (pre vs. post; means are displayed in Table 8).

As shown before, the knowledge transfer rate had a high impact on the satisfaction level (F $[1,315]$ = 88.19; $p < .001$)–with lowest transfer rates leading to the highest means for satisfaction (quartile 1 = 5.20, 2 = 4.46, 3 = 3.76, and quartile 4 = 3.26; see Table 8). The data show a significant difference in work satisfaction along the time scale (pre = 4.04 vs. post = 4.76; F

**Table 8. Means (and standard deviations) of work satisfaction (1 = lowest to 10 = highest satisfaction) as a function of time (pre, post), experience level (low vs. high), and knowledge transfer (quartile 1 to 4).**

| Point in Time | Prior experience | Knowledge transfer (quartile groups): Mean (Std. Dev.) | | | |
|---|---|---|---|---|---|
| | | 1: lowest | 2: below average | 3: above average | 4: highest |
| Pre | No 4.01 (1.44) | 4.93 (1.42) | 4.53 (1.28) | 3.59 (1.12) | 3.26 (1.29) |
| 4.04 (1.45) | Yes 3.98 (1.68) | 7.80 (1.00) | 4.80 (1.35) | 2.77 (0.28) | 2.68 (0.49) |
| Post | No 4.24 (1.73) | 5.41 (1.85) | 4.23 (1.73) | 4.00 (1.32) | 3.19 (1.09) |
| 4.76 (1.77) | Yes 5.27 (2.11) | 7.30 (1.41) | 4.14 (1.62) | 4.57 (1.81) | 4.26 (2.31) |
| Overall mean | | 5.20 (1.65) | 4.46 (1.43) | 3.76 (1.26) | 3.26 (1.29) |

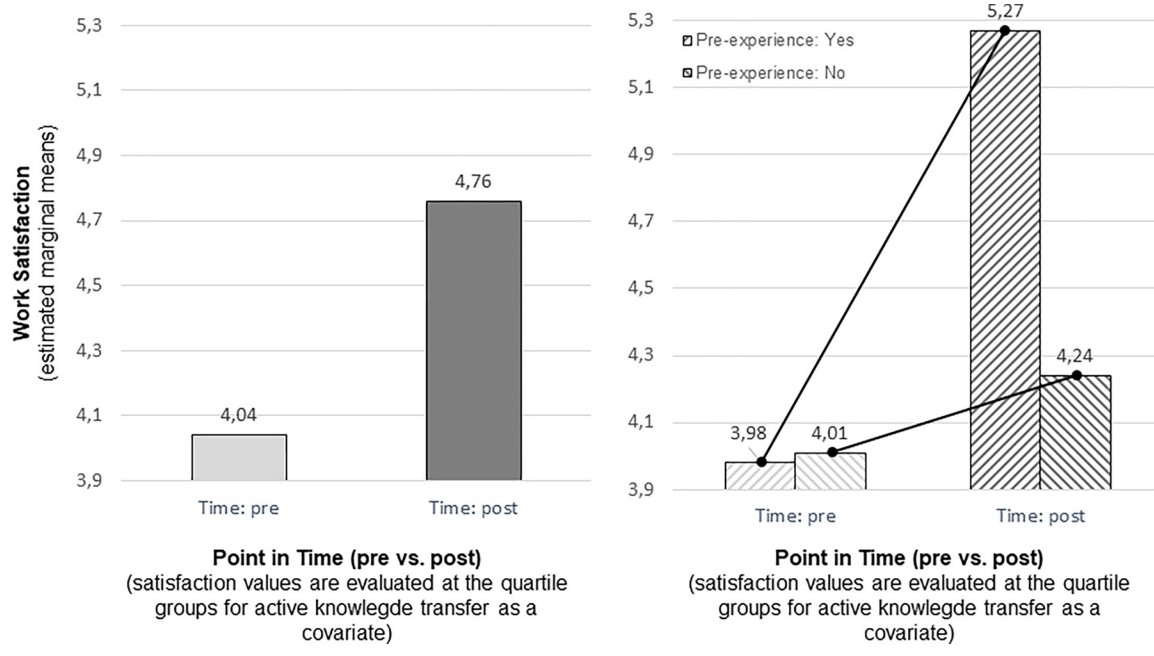

**Fig 4.** Impact of point in time (left) and interaction of time and experience (right) on satisfaction.

[1,315] = 6.51; p < .01; see Fig 4, left-hand side) when the influence of the knowledge transfer rate has been eliminated. As hypothesized, experience moderated this effect as work satisfaction increased for the more experienced employees (5.27 vs. 3.98), but not for less experienced ones (4.01 vs. 4.24; F[1,315] = 4,09; p < .05; Fig 4, right-hand side).

## Discussion

The surprising finding that emergency personnel with relatively low work satisfaction are more likely to share information with other staff members in times of digital transformation may be explained either by a general intensified level of talking and complaining due to poor satisfaction or by a higher level of meta-communication, that is, a shared reflection of the way in which medical staff is communicating and sharing knowledge [87]. Both explanations relate to making sense of an unclear or unknown situation, but in different ways: a generally higher level of communication helps to build trust and social orientation [88], whereas a higher level of meta-communication also helps to improve parameters for communication and knowledge sharing [87]. During the Rural|Rescue project an organizational change and a digital transformation process took place. Therefore, employees were exposed to rather high amounts of uncertainty or even anxiety [89]. Naturally, employees try to overcome uncertainty and help understanding changing workflows using "a series of interdependent social psychological properties" [90]. That means, they try to make sense of an environment they temporarily feel uncertain and anxious of. According to Watzlawick et al. [91], employees start to feel that their previously well-used references are obsolete. They react by altering their behavior in order to fit into the new situation.

### Emotion-based coping: Intensified communication

Intensive communication contributes to the process of sense-making, as various colleagues are likely to be in the same situation and mindset. If employees are not satisfied with their work

conditions, they exchange their point of view on an emotional basis. This intensive talking may not only lead to an increased perception of companionship, but also helps make sense of not well-understood formal communication in the organization or other situations that are responsible for low job satisfaction [92]. Informal communication, even including rumors has the potential to preempt formal communication from the organization itself, which establishes an informal sense-making process [93]. This can influence organizational activities in a positive way as information flows within the organization more easily [89] and, in return, may increase the employees' job satisfaction. Moreover, mechanisms like gossip directed against other staff members may also serve as a coping mechanism to overcome stressful situations and must be considered and mitigated when initiating organizational changes. The informal, emotion-based coping-mechanism of intensive, mostly informal talking, may increase job satisfaction—although this is not directly controllable by the management. The limited data available for this examination points in this direction but does not allow us to check for the presence of this coping mechanism.

**Content-based coping: Level of expertise.** According to Dunning [94], people have limited self-assessment and either over- or underestimate their knowledge depending on their level of expertise. Persons with pre-existing knowledge of the matter often underestimate their actual skill level [95]. Highly knowledgeable employees have a more critical attitude towards their competence level and feel the urgent need to fill possible knowledge gaps they notice [96]. This may also apply for the digital transformation we examined in the emergency patient care sector. Emergency personnel, who did not already have experience in using digital solutions like TEPs, might felt under pressure before and while using the digital solution for the first few times. They know that in hectic and stressful moments, it is crucial to use TEPs correctly. Uncertainties may already occur in pre-evaluation, because highly knowledgeable employees are aware of what is about to be implemented. But it is far more prevalent after implementation, when employees actively use the new digital feature and experience the discrepancy between their expectation and their performance regarding the usage of the digital tool in the beginning. This discrepancy can decrease employees' work satisfaction. To handle low work satisfaction due to perceived discrepancy, employees actively share their knowledge regarding TEPs in order to close the TEP knowledge gap and improving their performance. These sharing activities therefore are more content based.

The hypothesis, proposing a Dunning-Kruger effect, is supported, because higher TEP experience levels in the beginning affect the way satisfaction develops during organizational change: TEP-experienced employees, who were not content with their work conditions before digital transformation, become satisfied more easily as they are likely to close their TEP knowledge gap more quickly compared to less TEP-experienced employees with low work satisfaction. Increasing high skill levels due to new experiences should enable employees to use TEP more quickly.

The data show that experience affects the relationship of work satisfaction and active knowledge transfer. Indirectly, this demotes the finding that less content staff members cope by communicating a lot and as a result become more satisfied with their work conditions. It rather seems that those who already are highly knowledgeable using TEP search for more information and therefore adapt to the new situation more quickly which in return leads to higher work satisfaction later.

**Further research.** As already mentioned in the introduction of this paper, it is hardly possible to systematically observe and evaluate the interplay between job satisfaction, technology acceptance and knowledge sharing in a relevant medical setting. Therefore, scientific research towards (digital) transitions in the health care sector are rare. This study contributed to the literature by adding the perspective of paramedics in Northern Germany, who were involved in

a digital transformation process. Our paper is able to contribute to the finding of identifying factors supporting employees to accept and carry digital transformation. Those factors are crucial to be known to ensure high quality patient care provision and employee retention. The list of those factors is non-exhaustive and should be further charged in upcoming research.

In our study, we found evidence for a possible Dunning-Kurger-Effect in a field where continuous high-quality provision is key and where errors and insufficient workflows should be avoided. After our results contradicted our hypotheses backed by the majority of studies in this field, we tried to make sense of the results by looking at the impact of experience levels. This resulted in finding evidence for Dunning-Kruger-Effect. Further research is necessary to demonstrate the existence of the Dunning-Kruger-effect in these circumstances. Even if the Dunning-Kruger-Effect cannot be certified, it is worth investigating why knowledge transfer is higher when job satisfaction is lower.

**Implications for practice.**   The project Rural|Rescue is among the first projects to realign emergency medicine in rural areas using information and communication technologies (ICTs) and it is the first large scale transformation project in Germany. Its implementation was successful and can therefore be seen as promising practical example for other emergency patient care areas wanting to reduce their therapy-free interval and planning to introduce a digital transformation. There is increasingly more experience and reliable research on the use of TEP systems, e.g. [97]. The Rural|Rescue project was primarily concerned with the transformation of the rescue system and thus the adaptation of organizational, political, legal, economic and social framework conditions. Johansen et al. [8] have described success characteristics for the course of successful transition processes in the healthcare system as the actual core of "transformation knowledge". We have refined this framework [9] and summarized our transformation knowledge in Table 9 (see [86] for a full account of all measures and technical adaptations made to the TEP system).

For every organization making the decision to implement such a concept, it is crucial to ensure a successful implementation within an existing workflow which is pivotal for ensuring continuous high quality patient provision. As the results of this study suggest that employees are an important variable for the projects' successful implementation, it is necessary to know for the organization how to approach possible complications concerning the employees' acceptance and support towards digital transformation and their mechanisms to cope with change.

Organizations have substantial influence on increasing the probability of knowledge sharing which has an impact on employees' coping mechanisms—as our study showed. Wang and Noe [98] verified that an appropriate amount of supervisory control correlates to a higher frequency of knowledge sharing among employees. In this context, companies benefit from relatively flat organizational structures [99]. Organizational support, moreover, helps to empower and encourage employees to share knowledge [38, 39, 41, 100]. This support manifests itself in fairness and trust towards the employees as well as in innovativeness and affiliation [101]. Moreover, organizations can take various actions to achieve higher self-efficacy, which in turn can lead to increased knowledge sharing [39, 102], e.g. providing relevant trainings, removing knowledge sharing barriers [103] or establishing informal settings [104] like regular meetings, coffee breaks and interaction channels [38]—especially in order to ensure sense-making for employees who are not satisfied with current working conditions and are trying to overcome friction caused by organizational and/or technological change. It is crucial to provide networks where employees can establish connections and links among themselves [105].

Another point worth considering when initiating organizational changes in the healthcare sector is the deployment of change agents. Change agents are employees or leaders of the organization who are willing to support and promote organizational change efforts explicitly or implicitly [106]. Those agents can provide specific knowledge or influence the employees'

**Table 9. Success factors of transition process in the project Rural|Rescue [structured according to 8].**

| Element of transformation management | Description of the success factor in the project country| Rescue |
|---|---|
| Selective participation and multi-actor dynamics | Establishment of a "round table" at the responsible ministry, invitation of the project management to the district, health insurance companies, medical providers<br>Addressing regional and national multipliers (e.g., invitation of the Federal Chancellor to the Week of Revival) |
| Problem structuring | Formulation of the concept as an integrative approach based on care goals (e.g., shortening of the therapy-free interval) instead of technical upgrade (e.g., introduction of the tele-emergency doctor application)<br>Step-by-step adjustments and extensions in exchange with various actors |
| Vision with high attractiveness | Emphasis on the basic idea: Future-proof reorientation of rescue services in rural areas using telemedicine<br>Conviction through proof of necessity (sample calculation to comply with the legally prescribed auxiliary deadlines)<br>Cost calculation for alternative solutions<br>Identification of feasibility and positive effects<br>Presentation and active promotion of project progress (e.g., in news broadcasts and daily newspapers) |
| Platform / network for joint exchange | Gradual expansion of the "round table" to various discussion and expert groups<br>Participation in and organization of regional and national conferences<br>Presentation of project results in specialist publications |
| Open to trial and error | Try out solutions in practice (e.g., training concepts)<br>Further development of the formats through feedback from practice and evaluation results<br>Share experiences and use "best practice" as an opportunity for project changes (e.g., adaptation of forms and printouts) |
| Learning | Use of existing knowledge and experience advantages (e.g., internships and exchange with existing TEP applications)<br>Regular discussion of interim results with stakeholders ("reflexive evaluation")<br>Ongoing involvement of the ethics committee, project management agency and funding wise provider |

intensive talking behavior deliberately in order to enhance knowledge sharing activities and employees' job satisfaction. They can also facilitate the digital transformation without great friction and with an intention of not interrupting continuous high quality patient provision. External change agents are also possible [107], although the internal option may be implemented more easily as employees do not need to build a new relationship of trust with the change agent.

This requires a high amount of leadership which systematically and proactively minimizes possible struggles of the employees. With the mechanisms and management tools that were already explained, management should ensure high levels of job satisfaction and improved information flow within the emergency care provision. This applies generally, but especially in times of organizational or digital transformation and disruption. The management in particular is responsible for ensuring high quality of performance by motivating its employees and by increasing the information flow during organizational change.

Those actions not only enhance the possibility of sharing valuable knowledge but can also contribute to higher job satisfaction in general. In times of increasing shortage in medical staff, the importance of this aspect can hardly be overestimated. TEP may have a crucial significance for ensuring high quality patient care provision, but paramedics are necessary for its successful implementation and usage. For paramedics, emergency physicians as well as potential applicants, TEP could increase the employer attractiveness and general interest in the job. From a

societal perspective, TEP holds great opportunities such as reduced therapy-free intervals and high-quality patient care while less medical staff has to be on site.

## Conclusion

Using TEPs improves the quality of patient care significantly as it counterbalances the shortage of emergency physicians and reduces the therapy-free interval. Employees whose workflow is affected by digital transformation experience a lot of stress and insecurities and therefore need to be confident with the new working conditions. For the employer, it is crucial that employees use the new application determinedly and actively share their acquired knowledge.

Our research illustrates that employees' workload has an impact on the intention of using digital applications. The higher the workload, the more they are ready to accept an ICT solution like the TEP system. Regarding active knowledge sharing, we see that employees with low work satisfaction are more likely to share their digital knowledge compared to employees with high work satisfaction. We interpret this counterintuitive finding as a phenomenon of the Dunning-Kruger effect. Highly knowledgeable employees regarding the use of a digital solution feel uncertain about the change a first, which translates into temporarily lower work satisfaction. They feel the urge to fill even small knowledge gaps which in return leads to higher work satisfaction. Practitioners need to acknowledge that digital change affects their employees' workflow and job satisfaction. During such times of transformation, employees need time and support to gather information and knowledge in order to cope with digitally changed tasks.

## Supporting information

**S1 Appendix. Questionnaire.** Questions concerning knowledge transfer and knowledge sources.
(DOCX)

**S2 Appendix. Factor structure of the scale for active knowledge transfer (preliminary and final version).**
(DOCX)

**S3 Appendix. Means (and standard deviation) of technology acceptance and knowledge transfer as a function of point in time (pre, post), satisfaction (low, high) and workload (low, high).**
(DOCX)

**S1 File. Written consent form.** English translation of the German original.
(PDF)

**S2 File. Machine readable questionnaire.** German original, English translation in S1 Appendix.
(PDF)

**S3 File. Survey data.** Complete Survey data (SPSS format, English translation from the German original).
(SAV)

## Author Contributions

**Conceptualization:** Joachim P. Hasebrook, Leonie Michalak, Dorothea Kohnen, Bibiana Metelmann, Camilla Metelmann, Peter Brinkrolf, Steffen Flessa, Klaus Hahnenkamp.

**Data curation:** Joachim P. Hasebrook, Leonie Michalak, Dorothea Kohnen, Bibiana Metelmann, Camilla Metelmann.

**Formal analysis:** Joachim P. Hasebrook.

**Funding acquisition:** Joachim P. Hasebrook, Peter Brinkrolf, Steffen Flessa, Klaus Hahnenkamp.

**Investigation:** Joachim P. Hasebrook.

**Methodology:** Joachim P. Hasebrook, Leonie Michalak, Dorothea Kohnen.

**Project administration:** Joachim P. Hasebrook.

**Supervision:** Peter Brinkrolf, Klaus Hahnenkamp.

**Validation:** Camilla Metelmann, Steffen Flessa.

**Writing – original draft:** Joachim P. Hasebrook, Leonie Michalak, Bibiana Metelmann.

**Writing – review & editing:** Joachim P. Hasebrook.

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
