## [Decision Letter · Decision Letter 0]

28 Jul 2022

PONE-D-22-04801Digital transition in rural emergency medicine: Impact of job satisfaction and workload on communication and technology acceptancePLOS ONE

Dear Dr. Hasebrook,

Thank you for submitting your manuscript to PLOS ONE. After careful consideration, we feel that it has merit but does not fully meet PLOS ONE’s publication criteria as it currently stands. Therefore, we invite you to submit a revised version of the manuscript that addresses the points raised during the review process. Can you please address the reviewer's concerns below?

We look forward to receiving your revised manuscript.

Kind regards,

Avanti Dey, PhD

Staff Editor

PLOS ONE

“This research has been funded by a €5.38 million grant from Germany’s Innovation Fund (01NVF16004). Ethical approval BB 044/17 has been granted from Ethics Commission University Medicine Greifswald.”

“All authors participated in the funded project.

This research has been funded by a €5.38 million grant from Germany’s Innovation Fund (Innovationsfonds des Gemeinsamen Bundesausschusses, G-BA) under number 01NVF16004.

URL: https://innovationsfonds.g-ba.de/projekte/neue-versorgungsformen/landrettung-zukunftsfeste-notfallmedizinische-neuausrichtung-eines-landkreis.63

The funding institution had no role in study design, data collection and analysis, decision to publish, or preparation of the manuscript.”

“All authors declare no competing interests.”

7. Your ethics statement should only appear in the Methods section of your manuscript. If your ethics statement is written in any section besides the Methods, please delete it from any other section.

Reviewers' comments:

Reviewer's Responses to Questions

**Comments to the Author**

1. Is the manuscript technically sound, and do the data support the conclusions?

Reviewer #1: Yes

2. Has the statistical analysis been performed appropriately and rigorously? 

Reviewer #1: No

3. Have the authors made all data underlying the findings in their manuscript fully available?

Reviewer #1: No

4. Is the manuscript presented in an intelligible fashion and written in standard English?

Reviewer #1: Yes

5. Review Comments to the Author

Reviewer #1: This is a potentially interesting study and the author(s) tried to address this research issue in rural emergency medicine. Specifically, the title “Digital transition in rural emergency medicine: Impact of job satisfaction and workload on communication and technology acceptance”. However, this manuscript does satisfy the rigor and precision required for the work. I think the most important reason is that this paper clarity in theoretical support in Introduction section and hereafter in the literature review as well as implications all look good and properly well-designed study. Some comments and suggestions for improvement are as follows.

In the abstract section: this sentence can be revised “Data were collected via machine readable questionnaires issued to 755 persons (411 pre, 344 post), of which 304 or 40.3% of these persons responded (194 pre, 115 post)”.

At page 7, in design section, “During the first phase of the study (pre), 411 questionnaires were issued over a period of three months, of which 194 completed and evaluable questionnaires were returned (response rate 47.2%).” Is there any proper reason of lower response? Although, the response rate for any given survey may be one of the least important factors in analysing your results. There is no "accepted" response rate. It is accurate to say the higher the better, though. But author(s) needs to provide the proper reason and justification on this issue. Telephone surveys historically had the highest response rates, but they were also the most expensive. Old fashioned snail mail surveys had the lowest response rates, but they were cheaper. Technology continues to change survey research methods. Moreover, your response rate may differ depending on your population and geographic area. So researcher goal is not a high response rate, but the representation of the population of interest and low non-response bias (your entire data collection instrument affects this: population of interest, sampling, survey type, questions, benefit for the respondent, etc). But in your case situation is entirely different. So, you need to provide justification from your context. I hope you understand the importance of response rate in study. Sometimes researcher easily ignore the importance of this. So don’t be.

One last thing. For example: “Based on extensive preliminary studies published elsewhere [54-56], we combined these aspects in a polarity profile, which includes questions to be scored from 1 (positive pole) to 10 (negative pole; see table 2). An example question concerning support from colleagues reads as follows: "For my work I get sufficient appreciation and support from my colleagues" (positive pole=1) or "My work is not appreciated and unnecessarily criticized by my colleagues" (negative pole=10). The scale is highly reliable with Cronbach’s Alpha = 0.91 in this and 0.78 in prior studies [56].” Based on the preliminary studies published elsewhere? Is there any justification from the literature? According to my point of view, it is not necessary you combine the variables into the same or similar scale since they are measuring different constructs. Once they have been appropriately sectionalized and the items in the different sections have been content validated, it's okay. However, the reliably coefficient of each of the sections in line with the various variables under investigation should be separately calculated and results of the data collected from the use of the questionnaire should be reported separately in line with the variables and research objectives. I believe the author(s) should first address these fundamental flaws.

6. PLOS authors have the option to publish the peer review history of their article (what does this mean?). If published, this will include your full peer review and any attached files.

Reviewer #1: No

---

## [Author Response · Author response to Decision Letter 0]

26 Sep 2022

We would like to thank the anonymous reviewers for their helpful comments and recommendations helping us to improve our manuscript!

All changes in the manuscript (tracked version) have been highlighted in yellow. In what follows, we respond to the specific comments of the reviewer, individually. We have highlighted the aspects of the reviewers comments we are referring to in our response.

Reviewers’ comments are shown in italics.

Our response is marked with an �

This is a potentially interesting study and the author(s) tried to address this research issue in rural emergency medicine. Specifically, the title “Digital transition in rural emergency medicine: Impact of job satisfaction and workload on communication and technology acceptance”. 

However, this manuscript does satisfy the rigor and precision required for the work. I think the most important reason is that this paper clarity in theoretical support in Introduction section and hereafter in the literature review as well as implications all look good and properly well-designed study. Some comments and suggestions for improvement are as follows.

In the abstract section: this sentence can be revised “Data were collected via machine readable questionnaires issued to 755 persons (411 pre, 344 post), of which 304 or 40.3% of these persons responded (194 pre, 115 post)”.

We have added this sentence in the abstract and adapted the abstract in order to make purpose and implications of the study clearer.

At page 7, in design section, “During the first phase of the study (pre), 411 questionnaires were issued over a period of three months, of which 194 completed and evaluable questionnaires were returned (response rate 47.2%).” Is there any proper reason of lower response? Although, the response rate for any given survey may be one of the least important factors in analysing your results. There is no "accepted" response rate. It is accurate to say the higher the better, though. But author(s) needs to provide the proper reason and justification on this issue. Telephone surveys historically had the highest response rates, but they were also the most expensive. Old fashioned snail mail surveys had the lowest response rates, but they were cheaper. Technology continues to change survey research methods. Moreover, your response rate may differ depending on your population and geographic area. So researcher goal is not a high response rate, but the representation of the population of interest and low non-response bias (your entire data collection instrument affects this: population of interest, sampling, survey type, questions, benefit for the respondent, etc). But in your case situation is entirely different. So, you need to provide justification from your context. I hope you understand the importance of response rate in study. Sometimes researcher easily ignore the importance of this. So don’t be.

We added an explanation about the response rate as well as a comparison to the response rates found in similar studies. We also discuss the issue of a possible non-response bias. We added an explanation why the response rate in the post-phase of the study is lower than in the pre-phase.

One last thing. For example: “Based on extensive preliminary studies published elsewhere [54-56], we combined these aspects in a polarity profile, which includes questions to be scored from 1 (positive pole) to 10 (negative pole; see table 2). An example question concerning support from colleagues reads as follows: "For my work I get sufficient appreciation and support from my colleagues" (positive pole=1) or "My work is not appreciated and unnecessarily criticized by my colleagues" (negative pole=10). The scale is highly reliable with Cronbach’s Alpha = 0.91 in this and 0.78 in prior studies [56].” Based on the preliminary studies published elsewhere? Is there any justification from the literature?

We added the exact references and clarified which reliability was found in what study.

According to my point of view, it is not necessary you combine the variables into the same or similar scale since they are measuring different constructs. Once they have been appropriately sectionalized and the items in the different sections have been content validated, it's okay. 

We calculated and stated reliability (Cronbach’s Alpha) of each section of the survey we used as a factor in the analyses in text.

However, the reliably coefficient of each of the sections in line with the various variables under investigation should be separately calculated and results of the data collected from the use of the questionnaire should be reported separately in line with the variables and research objectives. I believe the author(s) should first address these fundamental flaws.

We arranged the order, in which the results are presented and added 3 tables reporting the results in line with the hypotheses and the analyses testing them. We moved an additional table (breakdown of means) to appendix 3.

---

## [Decision Letter · Decision Letter 1]

27 Oct 2022

PONE-D-22-04801R1Digital transition in rural emergency medicine: Impact of job satisfaction and workload on communication and technology acceptancePLOS ONE

Dear Dr. Hasebrook,

Thank you for submitting your manuscript to PLOS ONE. After careful consideration, we feel that it has merit but does not fully meet PLOS ONE’s publication criteria as it currently stands. Therefore, we invite you to submit a revised version of the manuscript that addresses the points raised during the review process.

We look forward to receiving your revised manuscript.

Kind regards,

Muhammad Rafiq, Ph.D

Guest Editor

PLOS ONE

Additional Editor Comments:

Thank you for submitting your paper about a “Digital transition in rural emergency medicine: Impact of job satisfaction and workload on communication and technology acceptance” to PLOS ONE journal. It is clear that a great deal of effort went into this paper, and we appreciate you entrusting your work with us. In terms of procedure, I have sent your manuscript out to reviewer(s) who are expert on the topics that your study focuses on. I have also carefully read the manuscript myself and have added some additional suggestions.

As you can see, the reviewers have positive comments about your paper, including that it is a timely and relevant topic, and especially its added value in terms of shortcomings in the current study. Yet, he/she also sees some clear limitations in your paper and provides constructive feedback on how your paper might be improved. I believe the comments and suggestions are clear. I also provide general comments on your paper below, so I will not repeat reviewers’ comments here.

The concept of digital transition is undoubtedly exciting and of interest to readers. However, I suggest authors give a detail discussion on the future research directions. Specifically, the research has not explained the implications of the research either for further research, practice or society, it is better that at the end of the study there is a sub discussion about this.

In the current form, this study was not enough to explain the position of the results of the study to previous studies.

Regarding methodology, it is also need serious improvement, mainly on how the researcher/s get the results and more in-depth addressing for the validity issues. Using a well-known methods used by previous researchers would solve the problem.

As you can see, there are quite some serious challenges that you would need to improve upon. The reviewer(s) and myself have made an effort to provide you with clear and constructive directions, aimed at helping you further improve your work, which is an important goal of the editorial process at PLOS ONE. I look forward to receiving your revised manuscript, and I wish you all the best in rewriting it.

Reviewers' comments:

Reviewer's Responses to Questions

**Comments to the Author**

1. If the authors have adequately addressed your comments raised in a previous round of review and you feel that this manuscript is now acceptable for publication, you may indicate that here to bypass the “Comments to the Author” section, enter your conflict of interest statement in the “Confidential to Editor” section, and submit your "Accept" recommendation.

Reviewer #2: (No Response)

2. Is the manuscript technically sound, and do the data support the conclusions?

Reviewer #2: Yes

3. Has the statistical analysis been performed appropriately and rigorously? 

Reviewer #2: Yes

4. Have the authors made all data underlying the findings in their manuscript fully available?

Reviewer #2: Yes

5. Is the manuscript presented in an intelligible fashion and written in standard English?

Reviewer #2: Yes

6. Review Comments to the Author

Reviewer #2: The authors have put considerable efforts in the improvement of their manuscript. My few suggestions are mentioned below:

The introduction section should depict the need for this study. It must establish the rational of grouping these variables.

The authors should also provide details regarding the theories used to underpin the relationship.

There is a need for theoretical model of this study.

The authors should highlight the hypotheses in a way that those should correspond to the literature.

The authors should proofread the manuscript once again for any potential errors.

7. PLOS authors have the option to publish the peer review history of their article (what does this mean?). If published, this will include your full peer review and any attached files.

Reviewer #2: No

---

## [Author Response · Author response to Decision Letter 1]

10 Dec 2022

PONE-D-22-04801R1

Digital transition in rural emergency medicine: Impact of job satisfaction and workload on communication and technology acceptance

Rebuttal Letter

Dear reviewers, dear editors, dear Dr. Rafiq,

we are grateful for your helpful comments and the opportunity to improve our manuscript. In what follows, we go through your requirements, step by step and in line with the structure of the manuscript (that is, from “introduction” to “discussion”).

Sincerely,

Joachim Hasebrook (in the name of all authors)

- The introduction section should depict the need for this study and must establish the rational of grouping these variables

We added a paragraph about the need for this study, and why job satisfaction, workload and communication are essential. We also added theoretical background about these aspects and their interrelation.

- The authors should also provide details regarding the theories used to underpin the relationship

We added details about the theories underpinning the different factors and their relationship.

- There is a need for theoretical model of this study

We added a section about the “Research model and Hypotheses” explaining the theoretical model behind our research.

- The authors should highlight the hypotheses in a way that those should correspond to the literature.

In the same section, we derived our hypothesis from the theoretical model and explicitly stated which references are related to the different hypotheses.

- in-depth addressing for the validity issues

We added paragraphs in the section “Material” about each instrument (job satisfaction and demands, technology acceptance, knowledge sharing) with details not only about reliability but also validity of the instruments applied in our study.

- how the researcher/s get the results

We added a paragraph in the section “Procedure” to explain how the results were gathered with machine-readable surveys, on site.

- explain the position of the results of the study to previous studies

We added text passages and paragraphs in the section “Discussion” to lay out, where our results support and contradict previous studies.

- give a detail discussion on the future research directions. Specifically, the research has not explained the implications of the research

We added paragraphs to both sections, “Future Research” and “Practical Implications”, to explain future research directions and their implications.

- The authors should proofread the manuscript once again for any potential errors.

We asked a professional proofreading service to improve our manuscript.

- We recommend that you deposit your laboratory protocols in protocols.io

The complete data set has been uploaded as supplementary file (S3)

Thanks very much to the editor and the reviewer for the opportunity

to revise and improve our manuscript!

---

## [Decision Letter · Decision Letter 2]

12 Jan 2023

Digital transition in rural emergency medicine: Impact of job satisfaction and workload on communication and technology acceptance

PONE-D-22-04801R2

Dear Dr. Hasebrook,

We’re pleased to inform you that your manuscript has been judged scientifically suitable for publication and will be formally accepted for publication once it meets all outstanding technical requirements.

Kind regards,

Muhammad Rafiq, Ph.D

Guest Editor

PLOS ONE

Additional Editor Comments (optional):

Thank you for resubmitting your manuscript to Plos One. Once again, it is clear that a great deal of effort went into this revising your work and meeting the reviewers’ requests. In terms of procedure, I have sent your manuscript out to the reviewer, and I have also carefully read it again myself as a reviewer and editor.

As you can see, the reviewer is positive about your revised manuscript and feels that you have done a great job in working with their previous comments. I agree with this conclusion: you have obviously done a good job at improving your manuscript in many ways. I applaud you for this great effort in revising your manuscript. Because of this, I am optimistic about publishing your work in Plos One in the near future.

Reviewers' comments:

Reviewer's Responses to Questions

**Comments to the Author**

1. If the authors have adequately addressed your comments raised in a previous round of review and you feel that this manuscript is now acceptable for publication, you may indicate that here to bypass the “Comments to the Author” section, enter your conflict of interest statement in the “Confidential to Editor” section, and submit your "Accept" recommendation.

Reviewer #2: All comments have been addressed

2. Is the manuscript technically sound, and do the data support the conclusions?

Reviewer #2: Yes

3. Has the statistical analysis been performed appropriately and rigorously? 

Reviewer #2: Yes

4. Have the authors made all data underlying the findings in their manuscript fully available?

Reviewer #2: (No Response)

5. Is the manuscript presented in an intelligible fashion and written in standard English?

Reviewer #2: Yes

6. Review Comments to the Author

Reviewer #2: (No Response)

7. PLOS authors have the option to publish the peer review history of their article (what does this mean?). If published, this will include your full peer review and any attached files.

Reviewer #2: No

---

## [Editor Report · Acceptance letter]

16 Jan 2023

PONE-D-22-04801R2 

Digital transition in rural emergency medicine: Impact of job satisfaction and workload on communication and technology acceptance 

Dear Dr. Hasebrook:

I'm pleased to inform you that your manuscript has been deemed suitable for publication in PLOS ONE. Congratulations! Your manuscript is now with our production department. 

Kind regards, 

on behalf of

Dr. Muhammad Rafiq 

Guest Editor

PLOS ONE